# An Estimation of Daily PM2.5 Concentration in Thailand Using Satellite Data at 1-Kilometer Resolution

Suhaimee Buya [1,2,*] , Sasiporn Usanavasin [1,*] , Hideomi Gokon [2] and Jessada Karnjana [3]

1  School of Information, Computer and Communication Technology, Sirindhorn International Institute of Technology, Thammasat University, Pathum Thani 12120, Thailand
2  School of Knowledge Science, Japan Advanced Institute of Science and Technology, Nomi 923-1211, Japan; gokon@jaist.ac.jp
3  National Electronics and Computer Technology Center, National Science and Technology Development Agency, Pathum Thani 12120, Thailand; jessada.karnjana@nectec.or.th
*  Correspondence: suhaimee.buy@dome.tu.ac.th (S.B.); sasiporn.us@siit.tu.ac.th (S.U.)

**Abstract:** This study addresses the limited coverage of regulatory monitoring for particulate matter 2.5 microns or less in diameter (PM2.5) in Thailand due to the lack of ground station data by developing a model to estimate daily PM2.5 concentrations in small regions of Thailand using satellite data at a 1-km resolution. The study employs multiple linear regression and three machine learning models and finds that the random forest model performs the best for PM2.5 estimation over the period of 2011–2020. The model incorporates several factors such as Aerosol Optical Depth (AOD), Land Surface Temperature (LST), Normalized Difference Vegetation Index (NDVI), Elevation (EV), Week of the year (WOY), and year and applies them to the entire region of Thailand without relying on monitoring station data. Model performance is evaluated using the coefficient of determination ($R^2$) and root mean square error (RMSE), and the results indicate high accuracy for training ($R^2$: 0.95, RMSE: 5.58 μg/m$^3$), validation ($R^2$: 0.78, RMSE: 11.18 μg/m$^3$), and testing ($R^2$: 0.71, RMSE: 8.79 μg/m$^3$) data. These PM2.5 data can be used to analyze the short- and long-term effects of PM2.5 on population health and inform government policy decisions and effective mitigation strategies.

**Keywords:** PM2.5 estimation; satellite data; aerosol optical depth; machine learning; random forest; Thailand





## 1. Introduction

According to the World Health Organization (WHO), ambient air pollution causes approximately 6.7 million premature deaths globally, with particulate matter, ozone, nitrogen dioxide, sulfur dioxide, and other contaminants being some of the leading pollutants [1]. The most dangerous among them is particulate matter with an aerodynamic diameter of less than 2.5 μm (PM2.5). These particles can easily enter the lungs and become trapped in the lung's parenchyma, leading to inflammation and oxidative stress [2]. This can cause severe cardiovascular and respiratory diseases and even lung cancer. PM2.5 plays a critical role in air pollution, and environmental health and its impact on human health are of great concern.

PM2.5 has been associated with increased mortality and morbidity in several studies [3–5]. However, the coverage of ground-level PM2.5 monitoring sites is limited, which makes it challenging to capture the spatial variability of PM2.5 for exposure and epidemiological research. Researchers have increasingly used satellite-derived atmospheric aerosol optical depth (AOD) to address this challenge as a proxy for ground-level PM2.5 [6–10]. AOD measures the aerosol in the atmosphere and can serve as a proxy for surface PM2.5 [11]. Additionally, other factor variables, including meteorological factors, land use and cover, and time variables, are often included to improve the accuracy of the modeling [12]. These variables can explain seasonal variations and long-term trends in PM2.5 levels and indicate

potential PM2.5 sources and areas of concern [13]. Conversely, the importance of these factors varies among studies, and some analyses have found that satellite-derived AODs do not improve model performance [14]. However, the study in the Pearl River Delta (PRD) region southern coast of China demonstrates the usefulness of AOD-derived spatiotemporal concentrations in health calculations [15]. Therefore, the association between satellite data and PM2.5 in different locations must be considered.

Previous studies on the estimation of PM2.5 using satellite data have employed a variety of models, but most have chosen only one [16]. The six studies were done to compare model performance comprehensively with the Random Forest (RF) model showing a high coefficient of determination ($R^2$) in four studies, and the eXtreme Gradient Boosting (XGBoost) model showing a high $R^2$ in two studies [13,14,16–19]. However, it should be noted that the RF model performed similarly to the XGBoost model. Among the other Machine Learning (ML) models, Multiple Linear Regression (MLR) had the lowest accuracy. Despite this, MLR is still widely used for its simplicity and practicality. Estimating PM2.5 concentrations is challenging due to the numerous variables that can affect it. ML has become popular for solving complex problems because it can find and use multiple independent factors that impact the predicted variable [20].

Earlier research on estimating PM2.5 levels in Thailand using satellite data has been limited due to a scarcity of data from both ground stations and satellites. Two previous studies conducted in Thailand's Chiangmai and central regions estimated PM2.5 using MLR models with AOD (10 kilometers (km)), resulting in $R^2$ values of 0.77 and 0.49 when considering monitoring station meteorological parameters and 0.22 and 0.11 when not considering them [21,22]. However, these meteorological parameters do not cover small areas such 1 km, 3 km, and 10 km, limiting the accuracy of PM2.5 estimation. A review article on predicting ground PM2.5 concentration using satellite AOD found that MLR had the lowest $R^2$ accuracy compared to other models [16]. The low $R^2$ values suggest further examination into including covariates such as meteorological factors, land use, cover, and season variables in MLR models [23].

In this study, we aim to develop a method for estimating PM2.5 concentrations throughout Thailand using satellite data with a 1 km pixel resolution. Our approach seeks to overcome the limitation of ground-level PM2.5 monitoring by not relying on monitoring station factor variables. Instead, we begin with AOD as a base factor and then add other variables to improve accuracy in estimating PM2.5 levels in Thailand. Specifically, we have selected Land Surface Temperature (LST), Normalized Difference Vegetation Index (NDVI), and Elevation (EV) data to represent land use and cover, as well as year and week of the year (WOY) as time factors. All factor variables are applied at a 1 km pixel resolution throughout Thailand without the need for monitoring station data, which can be costly and not cover all areas of the country. We will use MLR as the standard regression model and other ML models such as RF, XGBoost, and Support Vector Machines (SVM) to compare their performance. The final model with the highest accuracy will be selected to estimate PM2.5 levels in Thailand.

Our study will serve as a reference for future satellite-based PM2.5 estimation studies and will aid in exposure assessment in health studies of the Thai population. Using satellite data to estimate PM2.5 concentrations at a high spatial resolution, our study can provide a more comprehensive understanding of the distribution of PM2.5 in Thailand, which can help inform policy and public health efforts to reduce exposure to harmful air pollutants.

## 2. Materials and Methods

### 2.1. PM2.5 Data and Area of Study

Thailand is a Southeast Asian country that borders the Andaman Sea and the Gulf of Thailand, with an approximate population of 70 million people and an area of 513,120 square kilometers. The Pollution Control Department (PCD) is a legally recognized government agency in Thailand that collects data on air pollution parameters from meteorological stations throughout the country. Bangkok's Air Quality and Noise Management

Division (BAQ) also operates ground stations for monitoring PM2.5 in Bangkok. The PCD and BAQ measure PM2.5 data using the same standard, the beta-ray attenuation method, which follows the United States Environmental Protection Agency (USEPA) reference method. Figure 1 presents PM2.5 data and the number of stations from PCD and stations for BAQ from 2011 to 2020.

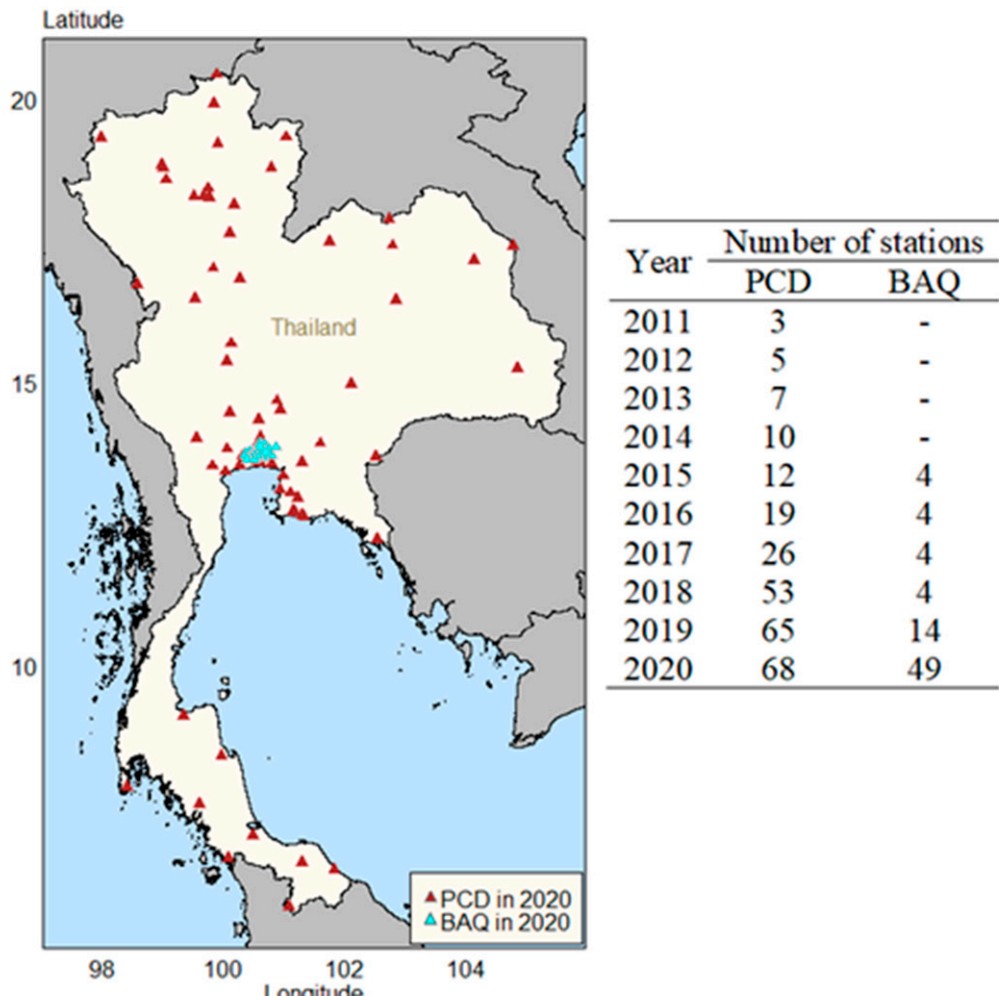

| Year | Number of stations | |
|---|---|---|
| | PCD | BAQ |
| 2011 | 3 | - |
| 2012 | 5 | - |
| 2013 | 7 | - |
| 2014 | 10 | - |
| 2015 | 12 | 4 |
| 2016 | 19 | 4 |
| 2017 | 26 | 4 |
| 2018 | 53 | 4 |
| 2019 | 65 | 14 |
| 2020 | 68 | 49 |

**Figure 1.** The map of PM2.5 stations and the number of stations.

### 2.2. Satellite Data

This study employed remote sensing data obtained from the MODIS satellite products, specifically AOD, LST, NDVI, and EV, which were all retrieved from the National Aeronautics and Space Administration (NASA) Earth Observing System Data and In-formation System (EOSDIS) offered by the Distributed Active Archive Center (DAAC). AOD data were processed from the MCD19A2 product of both Terra and Aqua satellites, which included "Aerosol Optical Depth at 045 Microns" [24]. The daily AOD data had a spatial resolution of 1 km per pixel and was collected at 10:30 a.m. and 1:30 p.m. local standard time. LST data was collected from Terra's MOD11A1 product and Aqua's MYD11A1 product [25], and their measurements were combined with increasing the sample size. Daily average LST values were calculated by taking the arithmetic mean of the two satellite measurements or using only one satellite's data. The study utilized NDVI data from MOD13A1, with a temporal resolution of 16 days and a spatial resolution of 500 m, which was beneficial in monitoring vegetation conditions, depicting land cover changes, and providing insights for modeling global biogeochemical and hydrologic processes and regional climates [26].

Additionally, EV data from "Land Digital Elevation Model (MODDEM1KM)—Land/sea mask and digital elevation model" with a spatial resolution of 1 km was used.

### 2.3. Data Analysis

For this study, we found that satellite data and PM2.5 readings were consistent when the sky was clear. At 1 km resolution, AOD and LST showed more than 50% missing values. However, the average over a 5 km radius only accounts for less than 50% of the missing number. To match the daily PM2.5 concentrations for each station from 2011 to 2020, we selected the average satellite data within a 5 km radius. We established a link between PM2.5 outcomes and factors such as AOD, LST, NDVI, EV, WOY, and year by using daily average PM2.5 data. Four models were developed to predict daily PM2.5: MLR, RF, XGBoost, and SVM. We evaluated the model's accuracy using $R^2$ and root mean square errors (RMSE). A higher $R^2$ and lower RMSE indicate better-estimating performance.

When extending this model estimation to other geographical areas, including regions, provinces, districts, and sub-districts, we can utilize the average satellite data within the boundaries of each specific area. Furthermore, data imputation techniques, such as nearest date and pixel, can be employed. The data handling and analysis procedures were implemented using the R programming language.

### 2.3.1. Multiple Linear Regression (MLR)

The MLR statistical model is a commonly used method for identifying the relationship between a continuous response variable and one or more predictor variables, which can be continuous or categorical. MLR is a parametric model that assumes a normal distribution, constant variance, and a linear relationship between the response and predictor variables. This study uses a log-linear regression model because the PM2.5 data has a skewed distribution, and the MLR model can be represented as:

$$\log(\text{PM2.5}) = \beta_0 + \beta_1 \text{AOD} + \beta_2 \text{LST} + \beta_3 \text{NDVI} + \beta_4 \text{EV} + \beta_5 \text{WOY} + \beta_6 \text{Year} \qquad (1)$$

where $\beta_0$ is the intercept, $\beta_{(1-6)}$ is the coefficient of determinant.

### 2.3.2. Random Forest (RF)

RF is a method for creating an ensemble of decision trees. The RF algorithm builds each tree using a bootstrap sample of the data, and each tree node is split based on the best of a subset of randomly selected predictors [27]. The predictions of each tree are then combined to produce an ensemble prediction of the target variable. The model also calculates the "importance" of each predictor by measuring how much prediction error increases when the data for that variable is permuted. In contrast, the data for the other variables remain unchanged [28]. This study uses the R package "randomForest" [29].

### 2.3.3. eXtreme Gradient Boosting (XGBoost)

XGBoost is a gradient-boosting technique that improves performance and speed using a tree-based ensemble ML algorithm [30]. Gradient boosting is a method where the loss function is minimized by sequentially adding weak learners through gradient descent optimization. The gradient boosting approach has three key components: a loss function, a weak learner, and an additive model. The loss function measures how well the model predicts the data. Even though a weak learner may not classify things accurately, it is still better than guessing randomly. The additive model is a method of adding decision trees one at a time and iteratively. This study uses the R package "xgboost" [31].

### 2.3.4. Support Vector Machines (SVM)

SVM is a supervised learning model for regression concerns in ML [32]. SVM builds a set of hyperplanes in a high-dimensional space using a nonlinear transformation based on the following function [33].

$$f(x) = wx + b \tag{2}$$

where x is the input predictors' vector (6 variables), w is the weight vector of x, and b is the error, which defines the hyperplane's distance from the original. SVM is based on decreasing the gap between the expected and actual output values. It reduces prediction errors. This study uses the R package "e1071" [34].

### 2.3.5. Model Assessment

The rows of the PCD dataset were randomly shuffled and divided into a training dataset (80%) and a validation dataset (20%) to ensure that model performance comparisons could be made. A consistent random state was used for this purpose. Table 1 presents the structure of the PCD and BAQ data. The distribution of the training and validation datasets were similar; however, the testing dataset was different as it only included BAQ data collected in Bangkok provinces.

**Table 1.** The data structure of datasets.

| Variables | Types | PCD (n = 34,748) | | BAQ (n = 7339) |
| --- | --- | --- | --- | --- |
| | | Training (n = 27,798) | Validation (n = 6950) | Testing |
| Stations | Nominal | 68 stations | 68 stations | 49 stations |
| Date | Date | 2778 days | 1865 days | 734 days |
| Month | Nominal | 12 months | 12 months | 12 months |
| Year | Discrete | 10 years | 10 years | 6 years |
| WOY | Nominal | 53 weeks | 53 weeks | 53 weeks |
| PM2.5 ($\mu g/m^3$) | Continuous | μ: 32.2, s: 23.7, IQR: 26 | μ: 32.4, s: 23.8, IQR: 26 | μ: 30.1, s: 16.2, IQR: 21 |
| AOD | Continuous | μ: 0.5, s: 0.3, IQR: 0.4 | μ: 0.5, s: 0.3, IQR: 0.4 | μ: 0.5, s: 0.3, IQR: 0.4 |
| LST (°C) | Continuous | μ: 33.3, s: 4.5, IQR: 6 | μ: 33.4, s: 4.5, IQR: 6 | μ: 36.1, s: 3.8, IQR: 4.3 |
| NDVI | Continuous | μ: 0.1, s: 0.2, IQR: 0.3 | μ: 0.1, s: 0.2, IQR: 0.3 | μ: −0.1, s: 0.1, IQR: 0.2 |
| EV (m) | Continuous | μ: 144.6, s: 198.9, IQR: 265.3 | μ: 142.4, s: 197.3, IQR: 265.3 | μ: 6.8, s: 1.6, IQR: 2.9 |

n: Rows; μ: Mean; s: Standard deviation; IQR: Interquartile range; m: Meter.

After training the model, the model's performance was evaluated by indicators such as $R^2$ and RMSE, shown in the following formulas:

$$R^2 = 1 - \frac{\sum (y_i - \hat{y}_i)^2}{\sum (y_i - \overline{y})^2} \tag{3}$$

$$RMSE = \sqrt{\frac{\sum (y_i - \hat{y}_i)^2}{n}} \tag{4}$$

where $y_i$ is the observations of PM2.5, $\hat{y}_i$ is the predicted value, $\overline{y}$ is the mean of the observations of PM2.5, and $n$ is the total sample count.

## 3. Results

### 3.1. Data Descriptive Statistics

Figure 2 presents a scatterplot matrix of the variables, with the first row and column displaying positive skew histograms of the PM2.5 distribution. Each scatterplot matrix includes the correlation coefficient (R) values, with the top row showing the relationship between each predictor variable and PM2.5. The first column displays the R values for all determinants with PM2.5. Positive R correlations between PM2.5 and AOD, LST,

and EV indicate that these variables increase along with PM2.5 (R = 0.51, 0.20, and 0.13, respectively), while negative R correlations between WOY (R = −0.27), NDVI (R = −0.19), and year (R = −0.05) and PM2.5 suggest that as these variables increase, PM2.5 will decrease. AOD has the highest positive association, and lower PM2.5 levels are observed during WOY 20-40 in Thailand's rainy season, indicating a negative correlation. Dry seasons with increased LST show higher PM2.5 levels, while higher NDVI levels decrease PM2.5. Finally, EV and Year have lower correlation values with PM2.5.

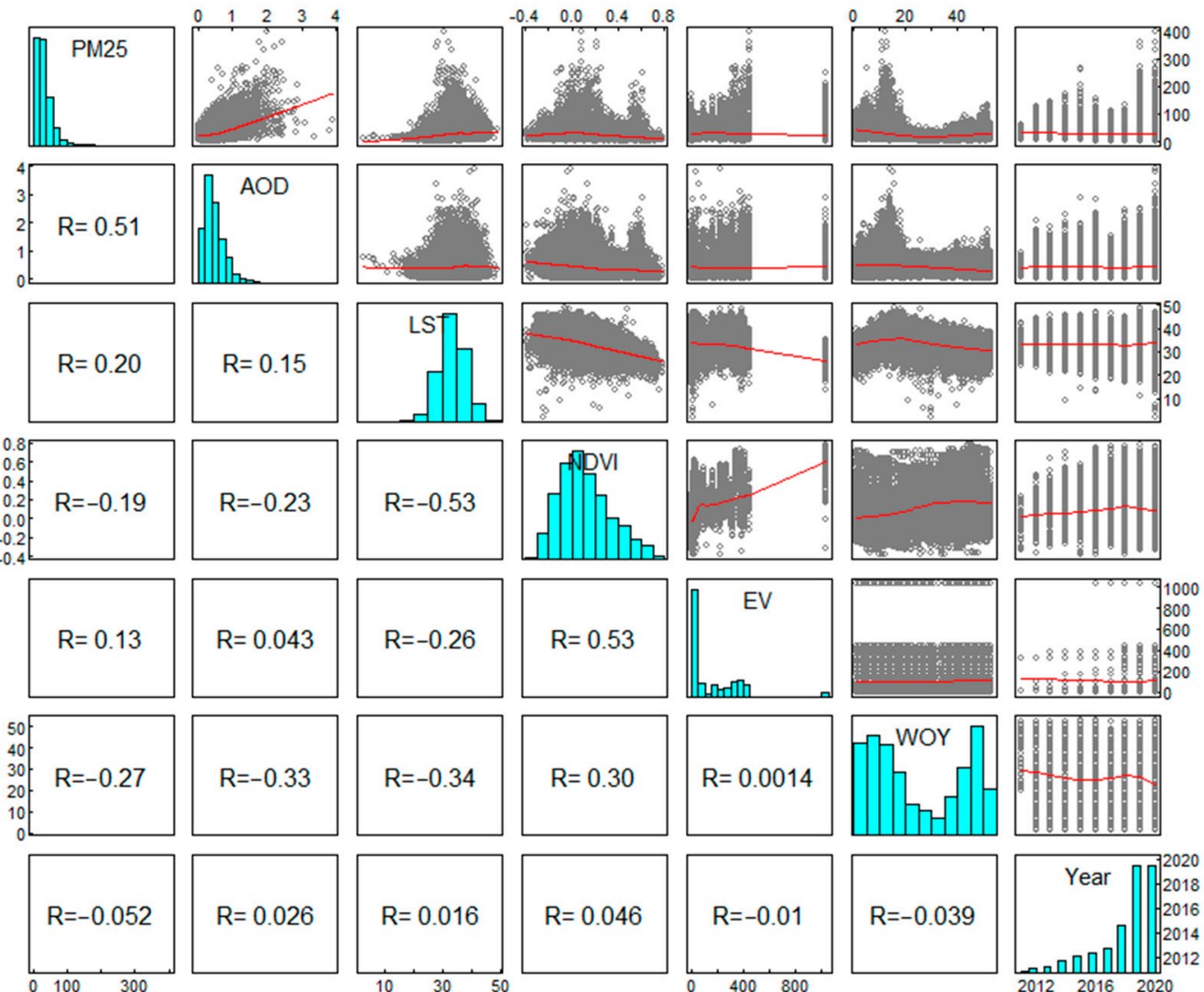

**Figure 2.** The scatterplot matrix of variables.

### 3.2. Modeling Results

Table 2 presents the estimated performance of each model for the three datasets. The results indicate that the RF model, which includes AOD, LST, NDVI, EV, WOY, and year, is the most effective in predicting PM2.5 across all datasets. The $R^2$ values for the training, validation, and testing datasets were 0.95, 0.78, and 0.71, respectively, with RMSE values of 5.58 μg/m³, 11.18 μg/m³, and 8.79 μg/m³, respectively. In terms of model performance, XGBoost and SVM were similar. However, the MLR model had the worst performance.

**Table 2.** The performance of models for estimation of PM2.5.

| Models | R² (RMSE (μg/m³)) | | |
|---|---|---|---|
| | Training | Validation | Testing |
| MLR | | | |
| AOD | 0.18 (21.48) | 0.19 (21.26) | 0.04 (16.79) |
| AOD + LST | 0.21 (21.25) | 0.22 (21.04) | 0.01 (17.15) |
| AOD + LST + NDVI | 0.22 (21.26) | 0.22 (21.19) | 0.01 (17.27) |
| AOD + LST + NDVI + EV | 0.25 (20.49) | 0.25 (20.38) | 0.01 (17.35) |
| AOD + LST + NDVI + EV + WOY | 0.51 (18.42) | 0.51 (17.94) | 0.35 (14.07) |
| AOD + LST + NDVI + EV + WOY + Year | 0.51 (18.28) | 0.52 (17.83) | 0.35 (13.78) |
| RF | | | |
| AOD | 0.79 (11.39) | 0.16 (23.08) | 0.02 (20.52) |
| AOD + LST | 0.86 (10.12) | 0.25 (20.88) | 0.04 (18.59) |
| AOD + LST + NDVI | 0.90 (8.82) | 0.44 (17.87) | 0.10 (16.03) |
| AOD + LST + NDVI + EV | 0.89 (8.82) | 0.60 (15.17) | 0.15 (15.05) |
| AOD + LST + NDVI + EV + WOY | 0.92 (7.23) | 0.74 (12.35) | 0.60 (10.47) |
| AOD + LST + NDVI + EV + WOY +Year | 0.95 (5.58) | 0.78 (11.18) | 0.71 (8.79) |
| XGBoost | | | |
| AOD | 0.31 (19.77) | 0.27 (20.27) | 0.04 (17.45) |
| AOD + LST | 0.34 (19.34) | 0.30 (19.85) | 0.05 (17.63) |
| AOD + LST + NDVI | 0.40 (18.39) | 0.38 (18.71) | 0.08 (15.90) |
| AOD + LST + NDVI + EV | 0.49 (16.94) | 0.47 (17.34) | 0.12 (15.23) |
| AOD + LST + NDVI + EV + WOY | 0.61 (14.93) | 0.60 (15.14) | 0.43 (12.40) |
| AOD + LST + NDVI + EV + WOY + Year | 0.62 (14.74) | 0.60 (15.00) | 0.45 (12.12) |
| SVM | | | |
| AOD | 0.28 (20.59) | 0.28 (20.66) | 0.04 (17.15) |
| AOD + LST | 0.31 (20.08) | 0.31 (20.16) | 0.05 (16.91) |
| AOD + LST + NDVI | 0.39 (18.83) | 0.38 (18.93) | 0.09 (15.68) |
| AOD + LST + NDVI + EV | 0.47 (17.60) | 0.46 (17.79) | 0.14 (15.65) |
| AOD + LST + NDVI + EV + WOY | 0.59 (15.64) | 0.60 (15.44) | 0.51 (11.51) |
| AOD + LST + NDVI + EV + WOY + Year | 0.61 (15.32) | 0.62 (15.17) | 0.52 (11.63) |

Although the final RF model has a higher $R^2$ accuracy in the validation dataset than the testing dataset, the testing dataset has a lower RMSE than the validation dataset. This means the RF model can estimate PM2.5 in the validation dataset more accurately than in the testing dataset. However, the difference between the actual and estimated PM2.5 in the testing dataset is closer than in the validation dataset due to the lower RMSE. This discrepancy could be attributed to the fact that the testing dataset only covers Bangkok provinces and thus has more data from these areas. In contrast, the validation dataset covers all areas of Thailand.

RF approaches were used to estimate daily PM2.5 concentrations in Thailand, and it was found that the model that included AOD, LST, NDVI, EV, WOY, and year had the best performance. The RF results also show two alternative measurements of each predictor variable's relative contribution in Figure 3. The %IncMSE is a percentage increase in mean square error, equivalent to accuracy-based importance. The IncNodePurity, calculated similarly to Gini-based importance, is based on reducing the sum of squared errors whenever a variable is split. Without WOY, AOD, EV, year, LST, and NDVI as predictors, the %IncMSE was 72.4%, 59.3%, 50.7%, 43.2%, 32.4%, and 31.5%, respectively. The important variables for IncNodePurity were WOY, AOD, EV, NDVI, LST, and year, respectively. These two measurements were calculated using different methods due to their strong association with ground-level PM2.5. Additionally, all the factors were needed to estimate PM2.5 levels in Thailand, where WOY, AOD, and EV were the three most essential variables in the two measurements.

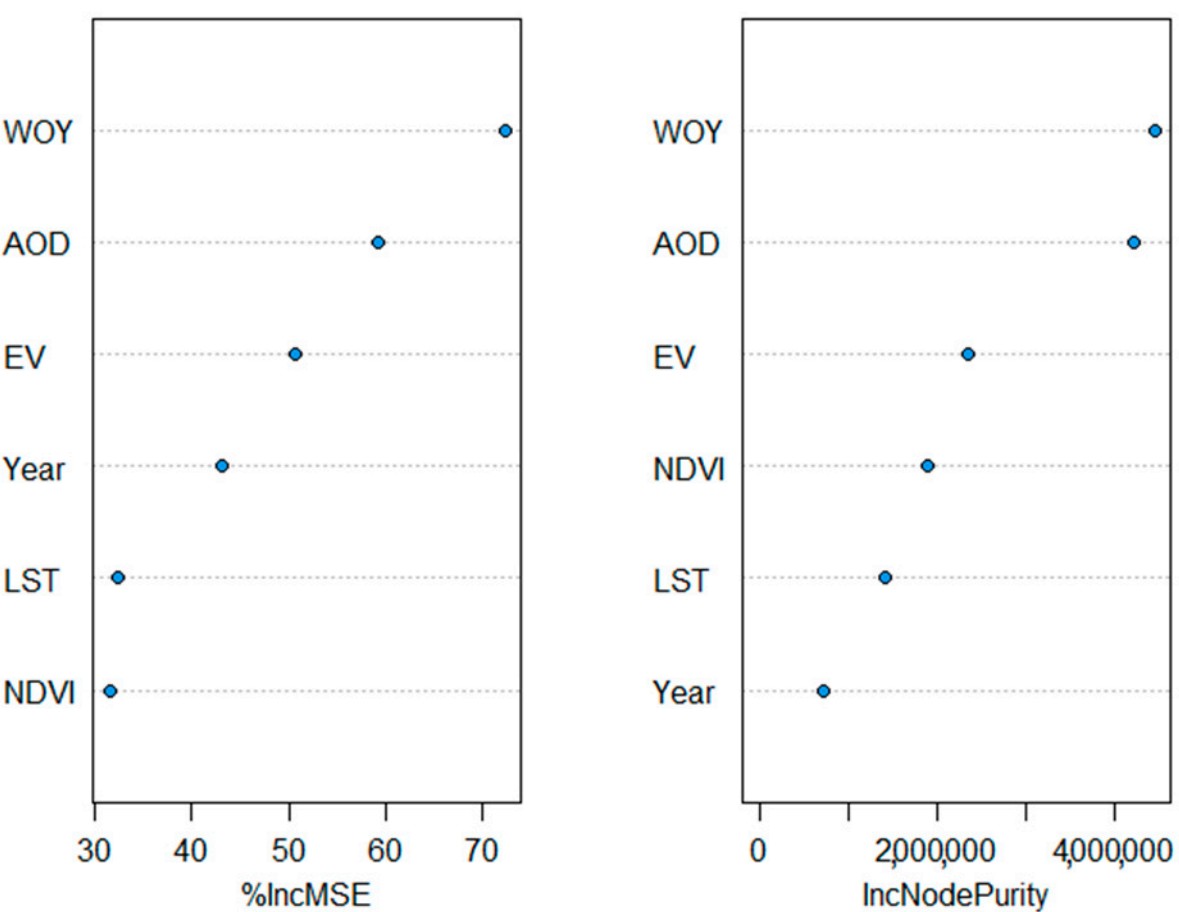

**Figure 3.** The importance variables for estimation of PM2.5.

### 3.3. Estimation of Daily PM2.5

Figure 4 presents the PM2.5 time series plot and estimation for the training, validation, and testing data. The three plots exhibit a consistent pattern in the observed and estimated PM2.5 concentrations, with the highest concentrations observed during weeks 45 to 53 (November to December) and 1 to 10 (January to March). The difference between the measured and estimated PM2.5 concentrations in the testing dataset was slight in 2015 and 2016 but remained consistent in 2017 and 2020.

Figure 5 presents the estimation of PM2.5 concentrations from 2011 to 2020 at a 1 km resolution using the RF model. The values of PM2.5 at stations and the estimated PM2.5 are comparable. The average percentages of correct estimation PM2.5 are between 68.9–75.2 with higher accuracy when PM2.5 is less than 15 μg/m$^3$ and higher than 50 μg/m$^3$. Northern Thailand exhibited the highest PM2.5 concentrations, while Southern Thailand showed the lowest levels. Except for the southern part of Thailand, most of the region's PM2.5 levels exceeded the WHO 24-h standard of 15 μg/m$^3$ but remained below Thailand's national standard limit of 50 μg/m$^3$ overall.

**Figure 4.** Time series plot of PM2.5 observed and estimation of PM2.5.

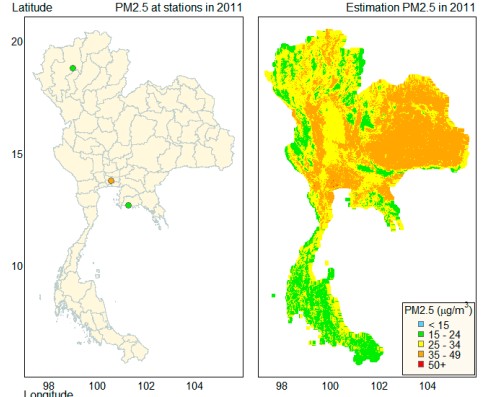

The percentages of correct estimation PM2.5 in 2011

| Estimation | Observed | | | | | Average |
|---|---|---|---|---|---|---|
| | <15 | 15–24 | 25–34 | 35–49 | 50+ | |
| <15 | 96.7 | 3.3 | 0 | 0 | 0 | **75.2** |
| 15–24 | 22.6 | 72.6 | 4.8 | 0 | 0 | |
| 25–34 | 0 | 22.7 | 68.2 | 9.1 | 0 | |
| 35–49 | 0 | 0 | 7.5 | 67.5 | 25 | |
| 50+ | 0 | 0 | 0 | 0 | 100 | |

**Figure 5.** *Cont.*

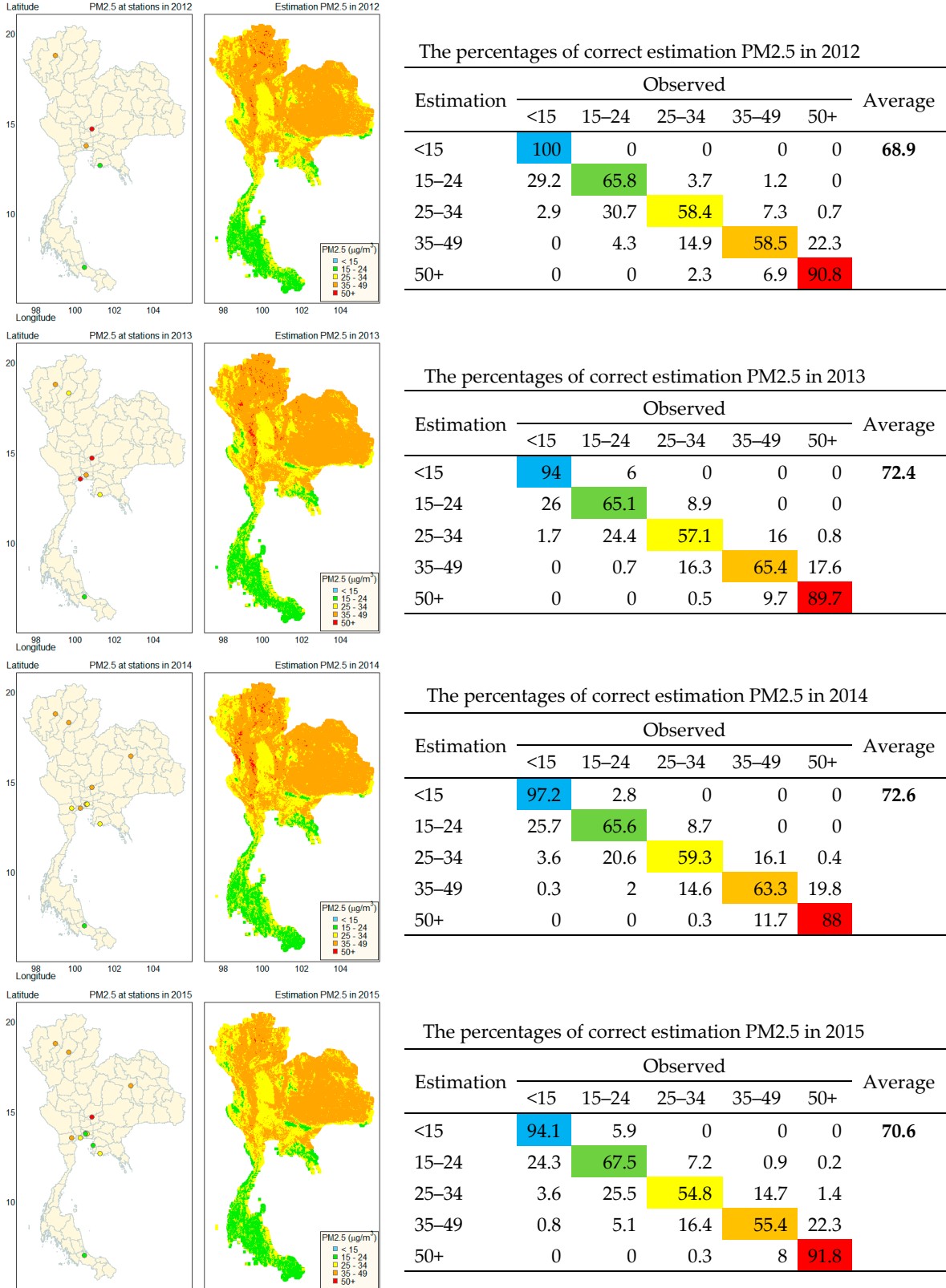

The percentages of correct estimation PM2.5 in 2012

| Estimation | Observed | | | | | Average |
|---|---|---|---|---|---|---|
| | <15 | 15–24 | 25–34 | 35–49 | 50+ | |
| <15 | 100 | 0 | 0 | 0 | 0 | **68.9** |
| 15–24 | 29.2 | 65.8 | 3.7 | 1.2 | 0 | |
| 25–34 | 2.9 | 30.7 | 58.4 | 7.3 | 0.7 | |
| 35–49 | 0 | 4.3 | 14.9 | 58.5 | 22.3 | |
| 50+ | 0 | 0 | 2.3 | 6.9 | 90.8 | |

The percentages of correct estimation PM2.5 in 2013

| Estimation | Observed | | | | | Average |
|---|---|---|---|---|---|---|
| | <15 | 15–24 | 25–34 | 35–49 | 50+ | |
| <15 | 94 | 6 | 0 | 0 | 0 | **72.4** |
| 15–24 | 26 | 65.1 | 8.9 | 0 | 0 | |
| 25–34 | 1.7 | 24.4 | 57.1 | 16 | 0.8 | |
| 35–49 | 0 | 0.7 | 16.3 | 65.4 | 17.6 | |
| 50+ | 0 | 0 | 0.5 | 9.7 | 89.7 | |

The percentages of correct estimation PM2.5 in 2014

| Estimation | Observed | | | | | Average |
|---|---|---|---|---|---|---|
| | <15 | 15–24 | 25–34 | 35–49 | 50+ | |
| <15 | 97.2 | 2.8 | 0 | 0 | 0 | **72.6** |
| 15–24 | 25.7 | 65.6 | 8.7 | 0 | 0 | |
| 25–34 | 3.6 | 20.6 | 59.3 | 16.1 | 0.4 | |
| 35–49 | 0.3 | 2 | 14.6 | 63.3 | 19.8 | |
| 50+ | 0 | 0 | 0.3 | 11.7 | 88 | |

The percentages of correct estimation PM2.5 in 2015

| Estimation | Observed | | | | | Average |
|---|---|---|---|---|---|---|
| | <15 | 15–24 | 25–34 | 35–49 | 50+ | |
| <15 | 94.1 | 5.9 | 0 | 0 | 0 | **70.6** |
| 15–24 | 24.3 | 67.5 | 7.2 | 0.9 | 0.2 | |
| 25–34 | 3.6 | 25.5 | 54.8 | 14.7 | 1.4 | |
| 35–49 | 0.8 | 5.1 | 16.4 | 55.4 | 22.3 | |
| 50+ | 0 | 0 | 0.3 | 8 | 91.8 | |

**Figure 5.** *Cont.*

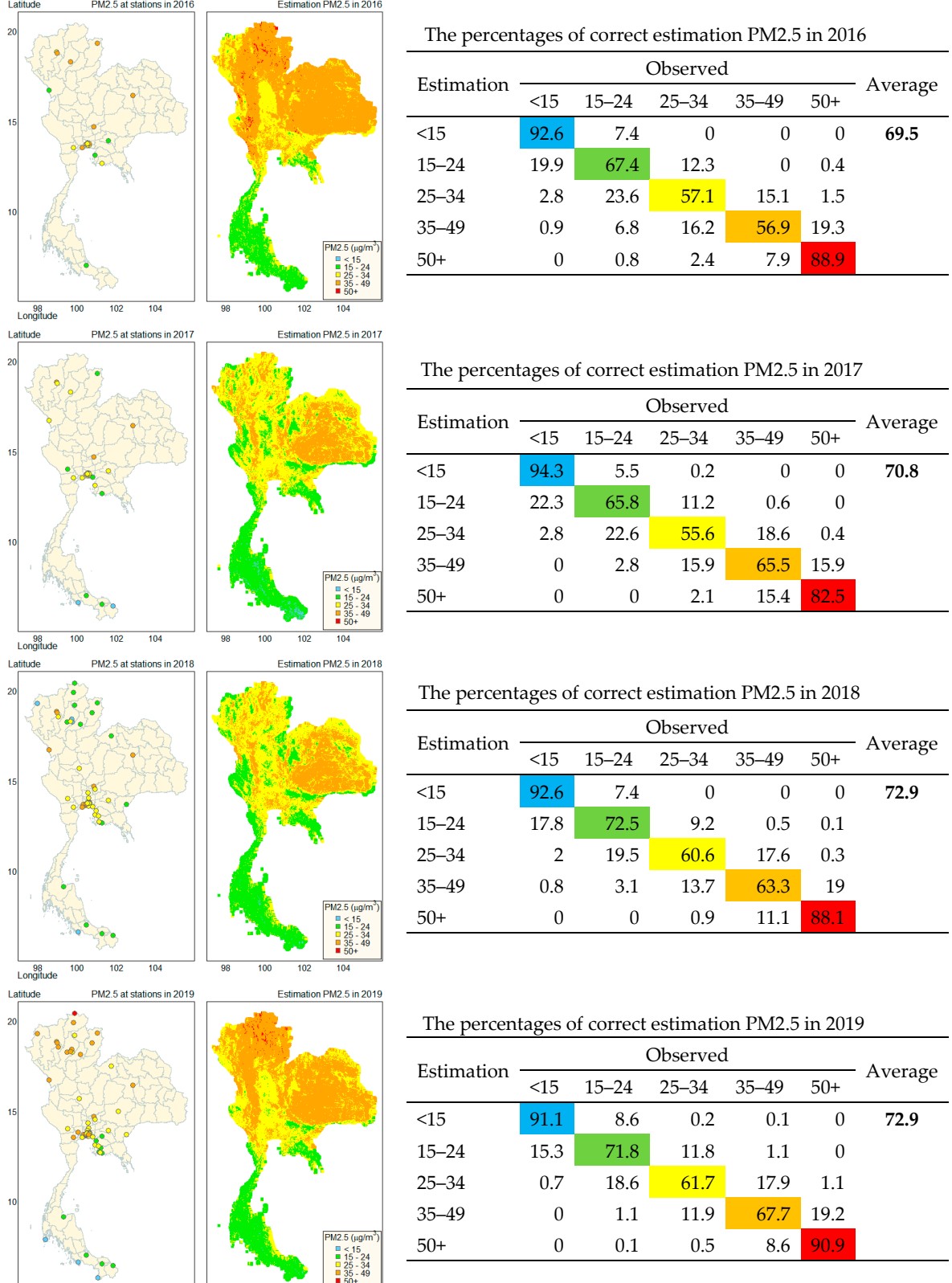

The percentages of correct estimation PM2.5 in 2016

| Estimation | Observed | | | | | Average |
|---|---|---|---|---|---|---|
| | <15 | 15–24 | 25–34 | 35–49 | 50+ | |
| <15 | 92.6 | 7.4 | 0 | 0 | 0 | **69.5** |
| 15–24 | 19.9 | 67.4 | 12.3 | 0 | 0.4 | |
| 25–34 | 2.8 | 23.6 | 57.1 | 15.1 | 1.5 | |
| 35–49 | 0.9 | 6.8 | 16.2 | 56.9 | 19.3 | |
| 50+ | 0 | 0.8 | 2.4 | 7.9 | 88.9 | |

The percentages of correct estimation PM2.5 in 2017

| Estimation | Observed | | | | | Average |
|---|---|---|---|---|---|---|
| | <15 | 15–24 | 25–34 | 35–49 | 50+ | |
| <15 | 94.3 | 5.5 | 0.2 | 0 | 0 | **70.8** |
| 15–24 | 22.3 | 65.8 | 11.2 | 0.6 | 0 | |
| 25–34 | 2.8 | 22.6 | 55.6 | 18.6 | 0.4 | |
| 35–49 | 0 | 2.8 | 15.9 | 65.5 | 15.9 | |
| 50+ | 0 | 0 | 2.1 | 15.4 | 82.5 | |

The percentages of correct estimation PM2.5 in 2018

| Estimation | Observed | | | | | Average |
|---|---|---|---|---|---|---|
| | <15 | 15–24 | 25–34 | 35–49 | 50+ | |
| <15 | 92.6 | 7.4 | 0 | 0 | 0 | **72.9** |
| 15–24 | 17.8 | 72.5 | 9.2 | 0.5 | 0.1 | |
| 25–34 | 2 | 19.5 | 60.6 | 17.6 | 0.3 | |
| 35–49 | 0.8 | 3.1 | 13.7 | 63.3 | 19 | |
| 50+ | 0 | 0 | 0.9 | 11.1 | 88.1 | |

The percentages of correct estimation PM2.5 in 2019

| Estimation | Observed | | | | | Average |
|---|---|---|---|---|---|---|
| | <15 | 15–24 | 25–34 | 35–49 | 50+ | |
| <15 | 91.1 | 8.6 | 0.2 | 0.1 | 0 | **72.9** |
| 15–24 | 15.3 | 71.8 | 11.8 | 1.1 | 0 | |
| 25–34 | 0.7 | 18.6 | 61.7 | 17.9 | 1.1 | |
| 35–49 | 0 | 1.1 | 11.9 | 67.7 | 19.2 | |
| 50+ | 0 | 0.1 | 0.5 | 8.6 | 90.9 | |

**Figure 5.** *Cont.*

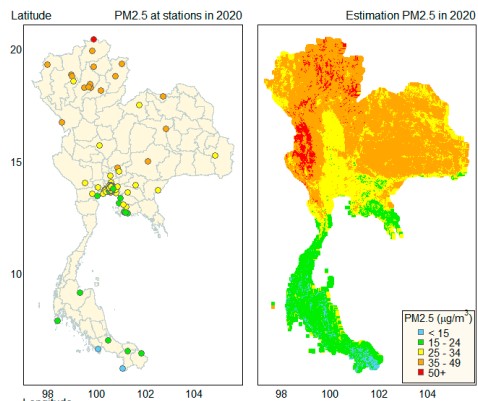

**Figure 5.** Estimation of PM2.5 in Thailand 2011–2020 in each pixel has a 1 km resolution.

| Estimation | Observed | | | | | Average |
|---|---|---|---|---|---|---|
| | <15 | 15–24 | 25–34 | 35–49 | 50+ | |
| <15 | 85.8 | 13.7 | 0.5 | 0 | 0 | **69.9** |
| 15–24 | 20.4 | 67.2 | 11.7 | 0.7 | 0.1 | |
| 25–34 | 2.2 | 23.2 | 55.2 | 18.7 | 0.7 | |
| 35–49 | 0.2 | 2.1 | 17.5 | 60.8 | 19.4 | |
| 50+ | 0 | 0.1 | 0.9 | 10.6 | 88.4 | |

The percentages of correct estimation PM2.5 in 2020

## 4. Discussion

We proposed using satellite data with a 1 km resolution to predict daily PM2.5 concentrations in Thailand and identified the best model to achieve this. The results of this model estimation can be utilized as standards for simulating PM2.5 in other areas with a similar mix of pollution sources and a need for more monitoring to understand the particle's spatiotemporal distribution. Investigating the spatiotemporal variations of PM2.5 at small scales was made possible by estimating PM2.5 in 1 km grid cells. These PM2.5 values are intended to aid epidemiological research and assist individuals in making informed decisions about air pollution.

In our trials, RF outperformed MLR, XGBoost, and SVM models. Our findings align with previous PM2.5 estimating studies from other countries, with an $R^2$ of 0.95 (RMSE of 5.58 $\mu g/m^3$) for training data, 0.78 (RMSE of 11.18 $\mu g/m^3$) for validation data, and 0.71 (RMSE of 8.79 $\mu g/m^3$) for testing data. For example, the predicted PM2.5 in Greater London using RF, Gradient Boosting Machine (GBM), and K-Nearest Neighbor (KNN), with RF providing the best estimation with an $R^2$ of 0.83 and RMSE of 4.28 $\mu g/m^3$ [35]. In another study, using remote sensing data and AOD, eight approaches were used to anticipate monthly PM2.5 in British Columbia, and RF was found to be the most reliable ML method, with an $R^2$ of 0.49 (RMSE of 2.67 $\mu g/m^3$) [18]. The predicted daily PM2.5 at a 1 km grid for 2013–2015 in Italy using RF with an $R^2$ of 0.80 (RMSE = 7.05 $\mu g/m^3$) [36]. The computed 1 km-resolution PM2.5 concentrations in China using RF, with an $R^2$ of 0.98 (RMSE = 6.40 $\mu g/m^3$) for model fitting and an $R^2$ of 0.81 (RMSE = 17.91 $\mu g/m^3$) for model validation [20]. Another Chinese study used RF to predict daily PM2.5 from 2005 to 2016, with an $R^2$ of 0.77 (RMSE of 22 $\mu g/m^3$) [17]. These studies demonstrate that estimating PM2.5 from satellite data using the RF model with an $R^2$ of 0.49–0.83 (RMSE = 2.67–22 $\mu g/m^3$) in the validation data is acceptable. On the other hand, the MLR model performed poorly in this study. This may be due to the positively skewed and non-normally distributed nature of PM2.5 data, which may not be well suited for MLR models [37–39].

The study found that the RF model, utilizing AOD, LST, NDVI, EV, WOY, and year as predictors, produced the best results for estimating daily PM2.5 concentrations in Thailand. The strength of the RF model lies in its ability to avoid overfitting data by utilizing the strength of individual trees in the forest and their correlation. However, the results of our study differ from those of other studies, where other models, such as XGBoost, have been found to outperform RF [17]. This may be due to how these decision tree-based models take in and process training data. Our findings suggest that decision tree-based models are recommended for estimating PM2.5 using satellite data.

The results indicate that WOY, AOD, and EV are significant factors in determining PM2.5 concentrations, as shown by the two measurements of the RF model. This is consistent with previous studies, which found AOD and EV to contribute to PM2.5 modeling

significantly [18]. Daily PM2.5 concentrations often exhibit a favorable skewed distribution similar to AOD. Similar to the research conducted in China, the bivariate correlation analysis revealed that independent variables such as AOD strongly associate with PM2.5 [20]. Our results also show that the estimated PM2.5 concentrations align well with the observed values at monitoring stations, with similar patterns in the time-series plots for observed and estimated PM2.5. However, there was some discrepancy between observed and estimated PM2.5 concentrations in 2015–2016. This may be due to the less varied geographical distribution of pollutants in the PM2.5 sample taken before 2017, as suggested by research from the United Kingdom [35].

The PM2.5 assessment indicates that northern Thailand experiences higher levels of PM2.5 than other regions, particularly during the dry seasons of WOY 1–10 (January–March) and WOY 45–53 (November–December). This is attributed to extensive agricultural fields and open-air biomass burning in northern Thailand and neighboring countries [22]. These activities contribute to the elevated PM2.5 levels and also have a significant impact on climate change. Except for the southern region, most areas in Thailand surpass the WHOs 24-h standard of 15 $\mu g/m^3$ for PM2.5 levels, although they remain within the national limit of 50 $\mu g/m^3$. The high PM2.5 levels can negatively impact population health, including respiratory and cardiovascular diseases. Our model's PM2.5 data can be used to identify links between PM2.5 levels and specific geographic areas, such as provinces, districts, and sub-districts.

Although satellite data can provide higher coverage than ground monitoring stations for PM2.5 data, it often has lower temporal coverage due to lousy observation conditions such as clouds and fog. We used average satellite data within a 5 km radius of the stations to decrease missing values. In our analysis, we used 42,009 (or 33.6%) data points out of 124,846 valid data points. According to evaluate MODIS collection 6 AOD retrievals against ground sunphotometer observations over East Asia cloud cover or high surface reflectance can cause an average of 40% to 70% of satellite retrievals to go unrecovered [40]. Furthermore, Thailand's overcast or foggy weather can invalidate the satellite retrieval technique by reducing the sampling frequency of accessible satellite data. This issue has also been identified in a study conducted in China [8]. As a result, new monitoring methods with wider spatial coverage and fewer weather limitations should be developed. These strengths can be used as benchmarks when estimating ground-level PM2.5 or other air pollution metrics in Thailand or other countries using remote sensing.

## 5. Conclusions

This study proposed an efficient method for estimating daily PM2.5 concentrations in Thailand using satellite data with a pixel resolution of 1 km. The RF model was the most effective compared to MLR, XGBoost, and SVM models. The use of AOD, LST, NDVI, EV, WOY, and year as predictor variables improved the model's performance, resulting in $R^2$ values of 0.95 (RMSE of 5.58 $\mu g/m^3$) for the training dataset, 0.78 (RMSE of 11.18 $\mu g/m^3$) for the validation dataset, and 0.71 (RMSE of 8.79 $\mu g/m^3$) for the testing dataset. The results from 2011 to 2020 were consistent with PM2.5 values obtained from monitoring stations. Using satellite data in this study allowed for examining air quality at various regional and temporal scales. The developed models and projections can aid regulatory operations and future epidemiological research in Thailand.

**Author Contributions:** S.B., Conceptualization, Formal analysis, Writing—original draft. S.U., Supervision, Writing—review & editing. H.G., Writing—review & editing. J.K., Writing—review & editing. All authors have read and agreed to the published version of the manuscript.

**Funding:** This study was encouraged by the Sirindhorn International Institute of Technology (SIIT), Thammasat University Research Fund and Japan Advanced Institute of Science and Technology (JAIST), and the research fund of Thailand's National Electronics and Computer Technology Centre (NECTEC).

**Institutional Review Board Statement:** Not applicable.

**Informed Consent Statement:** Not applicable.

**Data Availability Statement:** PM2.5 data from PCD (http://air4thai.pcd.go.th/webV2/history/, accessed on 18 May 2023) and BAQ (https://bangkokairquality.com/bma/report?lang=en, accessed on 18 May 2023). The satellite data can be assessed at (https://ladsweb.modaps.eosdis.nasa.gov/search/, accessed on 18 May 2023).

**Acknowledgments:** The Pollution Control Department and Bangkok's Air Quality and Noise Management Division provided the PM2.5 data, which the authors are thankful for. We appreciate Professor Don McNeil's wise counsel. We are also grateful to SIIT, Thammasat University and JAIST for thesis support for our research.

**Conflicts of Interest:** The authors declare no conflict of interest.

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
