# Peer review of "An Estimation of Daily PM2.5 Concentration in Thailand Using Satellite Data at 1-Kilometer Resolution"

_sustainability, doi:10.3390/su151310024_

Round 1

Reviewer 1 Report

This manuscript applied regression and machine learning methods to generate the daily PM2.5 concentrations in Thailand. Generally, the manuscript is well-written and shows promise. However, certain specific details in the content are not sufficiently clear. Please see the following details.

1. Line 51: Please revise the wording as follows: Satellite-derived AODs do not enhance the model's performance. However, please refer to the literature provided below, which demonstrates the usefulness of AOD-derived spatiotemporal concentrations in health calculations.

Lin, C., Li, Y., Lau, A. K., Deng, X., Tim, K. T., Fung, J. C., ... & Yu, Q. (2016). Estimation of long-term population exposure to PM2. 5 for dense urban areas using 1-km MODIS data. Remote sensing of environment179, 13-22.

2. Line 131-132: Please clearly indicate how the authors handle the data on cloudy days.

3. Figure 4: please adjust the x-axis. The numbers are confusing. Please indicate which stations were selected here?

4. The color bar in Figure 5 covers a large range, making it difficult to distinguish the gap between the observed (OBS) values and the estimated values. If possible, please provide a statistical performance matrix table to demonstrate the performance of the estimated values.

5. Please double check the format of the references and follow the journal requirement.

Minor editing is suggested.

Author Response

The responses to the reviewer are on the 1-2 page—the revised mark as Track changes and red text. The revised paper is based on 3 reviewer comments.

Reviewer 2 Report

1.      The research has several technical faults and mistakes, such as:

·        most phrases are very lengthy and should be trimmed.

·        The numbering style for references should be consistent, as seen in line 55.

·        The caption at line 109 should be changed, as should the editing error at line 157.

·        Why are capital letters used in the middle of some words?

·        Figure 2 needs to be revised.

·        Some references, such as reference number 3, "WHO. Ambient (outdoor) air pollution. World Health Organisation Geneva," should be completely given.

2.      The work's uniqueness, as well as its interpretation and application, are not sufficiently evident.

3.      There is insufficient survey in the literature and current work relevant to the research.

Author Response

The responses to the reviewer are on the first page. The revised mark as Track changes and red text. The revised paper is based on 3 reviewer comments.

Reviewer 3 Report

This paper is interesting and could be accepted after some revisions.

1.you only need define PM2.5 once.

2. In introduction section, you should state the status of PM2.5 pollution in Tailand.

3.Please explain what you mean in figure 2,  I feel confused.

4. the language should be polished.

the language should be polished.

Author Response

(The authors gave the same response as above.)

Round 2

Reviewer 2 Report

The authors well addressed the reply to the reviewer’s comments and revised their work accordingly. Therefore, I recommend the publication of the manuscript in its present form in Sustainability.